# New gravity-derived bathymetry for the Thwaites, Crosson and Dotson ice shelves revealing two ice shelf populations

Tom A. Jordan[1], David Porter[2], Kirsty Tinto[2], Romain Millan[3], Atsuhiro Muto[4], Kelly Hogan[1], Robert D. Larter[1], Alastair G.C. Graham[5], John D. Paden[6]

[1] British Antarctic Survey, High Cross, Madingley Road, Cambridge, CB3 0ET, UK
[2] Lamont Doherty Earth Observatory
[3] Institut des Géosciences de l'Environnement, Université Grenoble Alpes, CNRS, 38000 Grenoble, France
[4] Dept. of Earth and Environmental Science, Temple University, Philadelphia, PA 19122, USA
[5] College of Marine Science, University of South Florida, St Petersburg, FL 33701, USA.
[6] Center for Remote Sensing of Ice Sheets (CReSIS), The University of Kansas, Kansas 66045, USA

*Correspondence to*: Tom A. Jordan (tomj@bas.ac.uk)

**Abstract.** Ice shelves play a critical role in the long-term stability of ice sheets through their buttressing effect. The underlying bathymetry and cavity thickness are key inputs for modelling future ice sheet evolution. However, direct observation of sub-ice shelf bathymetry is time consuming, logistically risky, and in some areas simply not possible. Here we use new compilations of airborne and marine gravity, radar depth sounding and swath bathymetry to provide new estimates of sub-ice shelf bathymetry outboard of the rapidly changing West Antarctic Thwaites Glacier, and beneath the adjacent Dotson and Crosson Ice Shelves. This region is of especial interest as the low-lying inland reverse slope of the Thwaites Glacier system makes it vulnerable to marine ice sheet instability, with rapid grounding-line retreat observed since 1993 suggesting this process may be underway. Our results confirm a major marine channel >800 m deep extends tens of kilometres to the front of Thwaites Glacier, while the adjacent ice shelves are underlain by more complex bathymetry. Comparison of our new bathymetry with ice shelf draft reveals that ice shelves formed since 1993 comprise a distinct population where the draft conforms closely to the underlying bathymetry, unlike the older ice shelves which show a more uniform depth of the ice base. This indicates that despite rapid basal melting in some areas, these recently floated parts of the ice shelf are not yet in dynamic equilibrium with their retreated grounding line positions and the underlying ocean system, a factor which must be included in future models of this regions evolution.

## 1 Introduction

The Thwaites Glacier system is a globally important region of change in the cryosphere system (Fig. 1a) (Scambos et al., 2017). In this region the marine based West Antarctic Ice Sheet comes into direct contact with upwelling modified Circumpolar Deep Water (mCDW) which is warm relative to the typical cool dense shelf water on Antarctic continental shelves (Jenkins et al., 2018). This warm water can both erode the buttressing ice shelves, and directly melt the grounded ice, both factors driving dynamic thinning and retreat of glaciers and contributing to rising global sea level (Pritchard et al., 2012). The inland reverse

slope of the bed beneath Thwaites Glacier and some of the adjacent glaciers means that marine ice sheet instability may occur (Schoof, 2007;Weertman, 1974). In this case a feedback is setup where grounding line retreat exposes a progressively larger cross-sectional area of ice, hence more ice fluxes into the ocean leading to further glacial retreat. Satellite observations

revealing an increase in the velocity of ~100 ma$^{-1}$ extending ~100 km inland from the Thwaites grounding line and surface draw down of over 1 ma$^{-1}$ indicate that this region is changing now (Gardner et al., 2018;Milillo et al., 2019). It has been argued that dramatic retreat of the grounding lines of Thwaites, Pope, Smith and Kohler glaciers of between 10 and 30 km since 1993 (Fig. 1a) means that ice sheet collapse due to marine ice sheet instability may have begun (Rignot et al., 2014;Milillo et al., 2019;Joughin et al., 2014).

Understanding the bathymetry beneath the ice shelves separating the open marine realm of the Amundsen Sea Embayment and the grounded ice of the Thwaites Glacier system is of particular importance for the evolution of this region (Fig. 1a). The bathymetry beneath ice shelves is a fundamental control on the ice sheet stability as the shape of the water cavity is a first order control on sub-ice shelf currents, including the flux of warm, deep ocean water to the ice shelf bases and the grounding line beyond (Jacobs et al., 2011). Melting, thinning and ultimately disintegration of ice shelves will trigger faster glacial flow,

forcing glacial retreat, leading to global mean sea level rise (Scambos et al., 2004;Rignot et al., 2014). Cavity shape is also likely an important factor controlling the rate of melting close to the grounding line (Milillo et al., 2019;Schoof, 2007). Direct measurements of sub-ice-shelf bathymetry by seismic sounding is slow and often impractical due to the extremely crevassed environment (Brisbourne et al., 2014;Rosier et al., 2018). Exploration of sub-ice-shelf cavities using autonomous underwater vehicles can also be risky and time consuming to attain regional coverage (Jenkins et al., 2010;Davies et al., 2017). An

alternative technique to provide a first-order estimate of the bathymetry is the inversion of airborne gravity anomaly data, which can be collected quickly and efficiently over large areas.

Recovery of bathymetry from gravity data relies on the fundamental fact that the density contrast at the sea bed gives rise to significant and measurable gravity anomalies. A variety of techniques have been employed to invert gravity data for bathymetry. In the simplest case, the free-air anomaly is transformed directly to an equivalent surface assumed to reflect the

bathymetry. This can be done in 3D using a fast Fourier transform approach such as the Parker–Oldenburg iterative method (Gómez-Ortiz and Agarwal, 2005), as applied to the Larsen Ice Shelf (Cochran and Bell, 2012). Although the broad pattern of the bathymetry is resolved, transformation of gravity signals directly into equivalent topography can give rise to significant errors, attributed in the case of the Larsen Ice Shelf to geological factors such as crustal thickness and sedimentary basins distorting the gravity field (Brisbourne et al., 2014). An alternative technique models the bathymetry using gravity data along

multiple 2D profiles, for example across the Abbot Ice Shelf (Cochran et al., 2014) and outboard of Thwaites Glacier (Tinto and Bell, 2011;Tinto et al., 2011). Such models are constrained to match known topography and inferences about the underlying geology provide additional constraints. The 3D bathymetry beneath the Pine Island Glacier ice shelf was inverted from gravity data using a 3D prism model and a simulated annealing technique solving for bathymetry and a sedimentary layer (Muto et al., 2016). Although this technique returns a bathymetry model constrained by observations, it is not clear whether

signatures due to sediments and bathymetry can be reliably separated without *a priori* constraints such as seismic observations

(Roy et al., 2005). More recently, a 3D model constrained by regional bathymetry and subglacial topography was used to model bathymetry offshore of Pine Island and Thwaites glaciers, and beneath the Crosson and Dotson Ice Shelves (Millan et al., 2017). This model showed a complex topography with deep channels extending to the margin of the ice sheet, particularly in the Dotson-Crosson area where previously-unknown deep (>1200 m) channels were identified.

In this paper we re-evaluate the sub-ice-shelf bathymetry offshore from Thwaites Glacier and beneath the Crosson and Dotson Ice Shelves (Fig. 1a) through the integration of new airborne gravity data collected during the 2018/19 field season as part of the NERC/NSF International Thwaites Glacier Collaboration (ITGC), Operation IceBridge (OIB) (Cochran and Bell, 2010, updated 2018) and new marine gravity data from the R/V Nathaniel B. Palmer collected during the cruise NBP19-02 (Fig. 1b). To recover bathymetry from gravity beneath the ice shelves we employ an algorithm-based approach similar to that used for

the Brunt Ice Shelf (Hodgson et al., 2019). This approach constrains the recovered topography to match all direct topographic observations. This constraint helps account for geological factors such as variations in crustal thickness, sedimentary basins, or intrusions. We acknowledge that away from direct topographic observations the uncertainties in the bathymetric estimate due to geological factors increase. However, we suggest that using a well-constructed gravity-derived bathymetry is preferable to unconstrained interpolation across sub-ice-shelf bathymetric data gaps many 10s of kilometres wide. Such use of gravity

data is routine for predicting topography in un-surveyed parts of the ocean using satellite data (Smith and Sandwell, 1994) and is being used in the Arctic where higher resolution airborne data are included (Abulaitijiang et al., 2019).

Our results confirm the shape and position of the previously identified troughs (Millan et al., 2017). Differences in the inversion results beneath the inboard parts of some of the ice shelves are identified, reflecting the higher resolution of the new gravity data set and differences in the methods used. Our improved topographic estimate reveals variations in sub-ice-shelf cavity

thickness, which have implications for the rate at which the warm ocean water can access the present-day grounding lines, and the mechanism of grounding line retreat in these and other areas.

## 2 Methods

### 2.1 The integrated gravity and topographic data sets

We utilise airborne gravity data from OIB and the ITGC campaign, together with marine gravity data from cruise NBP19-02 (Fig. 1b and c). The OIB free-air gravity data were collected from a DC-8 aircraft travelling at ~120 ms$^{-1}$ at an altitude of ~450 m above the ice surface, using the Sander Geophysics AirGrav system (Studinger et al., 2008). These data have an error of ~1.67 mGal in this region and resolve anomalies with a ~10 km full wavelength (Cochran and Bell, 2010, updated 2018;Tinto and Bell, 2011). The ITGC campaign utilised a Twin Otter aircraft, flying at ~60 ms$^{-1}$, on average 340 m above the ice, and a

different 'strapdown' gravity approach based around an iMar Inertial Navigation System (INS) (Becker et al., 2015;Wei and Schwarz, 1998). The resulting data has an internal error from crossover analysis of 1.56 mGal and resolves wavelengths down to ~5 km (Jordan et al., 2020c) (see supplementary data Section S2 for details).

Airborne gravity data were restricted to lines flown at <1500 m above the surface. Of this subset over 95% of the data were collected at 450 m ±200 m above the surface. Upward and downward continuation of the gravity data to a common altitude was neglected as continuation by ~200 m will have little impact on the amplitude of the gravity anomalies (~1 mGal) given the ~1000 m range to the key bathymetric sources. Downward continuation can also introduce unnecessary artefacts and neglecting upward continuation preserves short wavelength gravity information. The data collected >650 m from the ice surface may give rise to an artificially smooth bathymetry, but are spatially restricted (SFig. 1), and do not appear to give rise to any anomalous signals in the integrated free air gravity dataset (Fig. 1c).

Marine gravity data from cruise NBP19-02 matched the pattern of the airborne anomalies, but were offset by ~7.14 mGal above the level of the airborne data. The majority of this offset is due to the difference between geoid (marine) and ellipsoid (airborne) references used for the different systems. In the area of overlap the geoid-ellipsoid difference results in a ~9 mGal discrepancy, based on the GOCO3s satellite gravity model (Pail et al., 2010). The residual ~ 2 mGal difference may reflect drift in the marine system, and potential discrepancies in base station ties between the different surveys. Alternatively un-considered shorter wavelength variability in the gravity field not resolved by the GOCO3s model (<~160 km), or temporal changes in the geoid associated with glacio-isostatic adjustment and mass loss may account for the residual shift. Such features do not impact the locally recovered bathymetry and are beyond the scope of this paper. The average measured shift of 7.14 mGal was therefore subtracted from the marine line data as a single DC value. All line data were then merged into a single database, interpolated onto a 1 km mesh raster and filtered with a 5 km low pass filter removing residual line to line noise. This filter wavelength is justified as anomalies with wavelengths <5 km are not resolved by the airborne gravity systems used. The final, integrated free-air gravity map (Fig. 1c) shows a clear pattern of high and low anomalies, which to first order match the main 5-10 km wavelength features visible in the available onshore subglacial topography and offshore bathymetry (Fig. 1d).

The topographic observations onshore were taken from OIB line radar data (Paden et al., 2010, updated 2018), augmented with new depth sounding radar collected along with the gravity data during the ITGC campaign (Fig. 1d). This new bed elevation data was collected using a 600-900 MHz accumulation radar provided by the Center for Remote Sensing of Ice Sheets (CReSIS). Bed elevations were picked from SAR processed radargrams in a semi-automated fashion. Although the primary target of this radar system was shallow ice sheet structures bed elevation was resolved through ice up to ~1900 m thick. Visual inspection revealed a few incorrect onshore bed picks in the OIB dataset on Bear Island, which gave bed elevations above the highly accurate REMA surface digital elevation model (DEM) (Howat et al., 2019). These points were deleted from the integrated line bed elevation dataset. The line bed elevation data were corrected to the GL04c Geoid (Forste et al., 2008), and the data interpolated onto a 1 km mesh raster. This gridded dataset was carefully masked to remove regions which are now covered by the floating ice shelf based on the most up to date grounding lines (Rignot et al., 2014;Milillo et al., 2019). Bed elevation values over local sub-shelf pinning points were also excluded. This masking mitigates the risk of the base of a floating ice shelf being misidentified as a bed elevation point and biasing the inversion. Beyond the ice shelves we took

the values constrained by a new compilation of shipborne multibeam swath bathymetric data (Hogan et al., 2020), which was down sampled to a 1 km mesh raster for this study (Fig. 1d).

## 2.2 Recovering sub-ice-shelf bathymetry

To recover a gravity-enhanced bathymetry we follow an algorithmic approach, rather than a pure inversion (Hodgson et al., 2019). We refer to this as the *topographic shift* method, as an initial topographic estimate derived from the gravity data is shifted to match observed topographic tie points. Summarising the method, the initial 3D topographic estimate (SFig. 2a) was calculated from the free-air anomaly (Fig. 1c) using an iterative forward modelling method (von Frese et al., 1981). Differences with between the initial topographic estimate (SFig. 2b), and the observed bathymetry and onshore topography were calculated (Fig. 1d) and interpolated using a tensioned spline (Smith and Wessel, 1990). This difference grid was then subtracted from the initial topographic estimate to provide the final bathymetric estimate (Fig. 2a). For full detail on the method see supplementary material Section S1. The topographic shift method is conceptually similar to the *gravity shift* method developed and applied along the Greenland coast where the initial free-air gravity data were shifted to match the variable gravity field from models of known topography prior to inversion for bathymetry (An et al., 2019). This gravity shift method was subsequently employed to fill the sub-ice shelf bathymetry in the Thwaites Glacier region of the recent BedMachine Antarctica compilation (SFig.5c) (Morlighem et al., 2020). The advantage of both the topographic and gravity shift techniques is that features in the gravity field due to variations in crustal thickness, sedimentary basins or intrusions are implicitly taken into account, as long as they overlap with the topographic control points. This assumption is most robust for long wavelength features such as variations in crustal thickness, or regional sedimentary basins, where the associated errors will impact multiple topographic control points allowing good control of the resulting error field. The impact of more localised geological features which only partially overlap constraining topographic data will be less well defined and we make the assumption that such errors fall off smoothly away from the affected control points. Geological features which have no overlap with constraining topographic observations can still introduce artefacts distorting the recovered bathymetry in proportion to their size and density contrast.

## 2.3 Ice shelf draft and cavity thickness

The depth of the ice shelf base, and the thickness of the sub-ice-shelf water filled cavity (Fig. 2b) were calculated assuming the ice shelf is in hydrostatic balance (Griggs and Bamber, 2011). Hydrostatically defined draft is typically a good approximation to radar measured ice thickness (Griggs and Bamber, 2011) and provides seamless coverage of our study area. The input surface elevation data were taken from the REMA digital elevation model (DEM) (Howat et al., 2019) corrected to the GL04c Geoid (Forste et al., 2008), and re-sampled onto a 500 m grid cell size raster. In the study area the REMA DEM is based on satellite observations between 2014 and 2016, and therefore reflects the surface elevation after widespread un-grounding between 1993 and 2014 (Rignot et al., 2014). Ice and water densities were assumed to be 917 and 1027 kg m$^{-3}$,

respectively, and a 16 m firn correction was applied (Griggs and Bamber, 2011). Uncertainties in these assumed values may have an impact on the precise values of ice shelf draft, but are unlikely to significantly distort the calculated pattern of water

cavity thickness. Comparison between ice shelf base calculated from the higher resolution DEM and the longer wavelength bathymetry resolved by the gravity (>5 km) will introduce high frequency features into the estimated cavity depth, and unresolved bathymetric features will generate errors in cavity thickness. However, the regional trends in cavity thickness will not be affected and can be discussed. Errors in ice shelf draft of up to 80 m in the 10-25 km most proximal to the grounding line may occur due to the rigidity of the ice shelf (Rignot et al., 2011), but in Thwaites glacier this 'bending zone' appears to

be narrow (<5 km) (Milillo et al., 2019), which we attribute to the highly fractured nature of the ice shelf in this region.

## 3 Results

Our modelled sub-ice-shelf bathymetry (Fig. 2a) reveals a complex offshore topography from ~250 to >1000 m deep, with a pattern of ridges and troughs of a size and scale consistent with the terrain mapped onshore with radar, and offshore with multi-beam swath bathymetry. All the key bathymetric features we observe are imaged as anomalies in the free-air gravity data, and

are therefore not artefacts of the inversion technique. Many of the isolated pinning points seaward of Thwaites Glacier and beneath the Crosson Ice Shelf shown by InSAR-derived grounding lines (Rignot et al., 2014) are revealed by our study as being situated on broader bathymetric highs. In these areas our recovered topography predicts that the ice shelf is grounded, or within 100 m of grounding (i.e., the water column is calculated to be less than 100 m thick; Fig. 2b). As our inversion did not use any additional data (swath, seismic or radar) to constrain the elevation at these isolated pinning points within the ice

shelves the fact that many appear to be within error of their grounding level provides qualitative support for the reliability of our inversion.

The revealed sub-ice-shelf cavity is >500 m thick in many areas. Adjacent to parts of Thwaites Glacier this deep cavity reaches to within 0-10 km of the grounding line. In contrast the inboard parts of the Dotson and Crosson Ice Shelves formed since 1993 overlie a cavity typically <150 m thick (Fig. 2b), and the thick (>450 m) cavity lies more than 10-30 km from the current

grounding line.

Profiles of the bathymetry beneath the ice shelves confirm the complex sub-ice-shelf pattern (Fig. 3). Our results show that the tips of both the Eastern Ice Shelf and Thwaites Glacier Tongue are grounded at their seaward ends on a linear but dissected ridge, while a ~1000 m deep sub-ice-shelf cavity is apparent behind the pinning ridge (Figs. 3a and b). Where the grounding line of the Thwaites Glacier Tongue has retreated since 1993 the estimated ice shelf base closely follows the modelled

bathymetry (Fig. 3b). Along the narrow channel close to Bear Island a cavity >500 m thick is apparent beneath the Crosson Ice Shelf, but this does not extend into the region where the grounding line has retreated most significantly in the recent decades (Khazendar et al., 2016) (Fig. 3c). Profiles across the Dotson Ice shelf towards Kohler Glacier indicate the grounding line is separated from the main sub-ice-shelf cavity by a sill, which appears to reach within ~200 m of the base of the ice shelf (Figs. 2b and 3d).

## 4 Discussion

### 4.1 Quantification of errors

Quantification of the errors associated with gravity inversions is challenging as a combination of intrinsic but quantifiable uncertainties in the gravity data, the inversion assumptions, and the poorly understood variability of sub-surface geology all contribute to the error budget. Errors in the gravity field of ~1.56 mGal defined from crossover analysis directly contribute to ~23 m uncertainty in the recovered bathymetry. The modelled rock density of 2670 kg m$^{-3}$ assumes no sediments are present at the sea floor. This is reasonable given the generally rugged morphology observed across many parts of the Amundsen Sea inner shelf (Nitsche et al., 2013;Graham et al., 2009;Larter et al., 2009). However, assuming all bathymetry was carved into lithified sediment the total amplitude of the sub-ice-shelf topography could be underestimated by ~11% (~130 m), assuming a typical sediment density of 2500 kg m$^{-3}$ (Telford et al., 1990). Lower density un-lithified sediment could lead to an even larger underestimates of topographic amplitude, but such material would not be expected to form all of the >1000 m high ridges recovered by our inversion and imaged by recent swath data (Hogan et al., 2020). Other geological factors such as dense gabbroic intrusions, or local sedimentary basins could further distort the recovered bathymetry if they are away from the direct bathymetric observations which would mitigate the impact of such features on the final bathymetric model. Underlying geological factors can, in some cases, be revealed by coincident aeromagnetic data, as in the case of the Brunt Ice Shelf (Jordan and Becker, 2018;Hodgson et al., 2019) and Ross Ice Shelf (Tinto et al., 2019). In our study, tight correlation between high amplitude magnetic (Jordan et al., 2020b) and gravity anomalies is only seen beneath the grounded part of Thwaites Glacier (Fig. 3b). Such tight correlation is indicative of a significant geological feature distorting the gravity signature (Jordan and Becker, 2018), but is not seen on profiles across the offshore regions (Fig. 3). This favours a model where underlying geological factors are not dominating the inversion results.

In addition to quantifying the errors it is important to note that the resolution of the bathymetry recovered from gravity data is limited by the wavelengths resolved by the gravity systems and the survey line spacing. For this study the gravity systems resolved minimum wavelengths of 5 to 10 km and a minimum line spacing of ~5 km is achieved outboard of Thwaites Glacier, while a minimum line spacing of ~7.5 km was achieved over the Dotson and Crosson Ice Shelves. This study therefore only recovers bathymetric features with a wavelength of ~5 km and upwards.

To best quantify the uncertainty in the sub-ice-shelf bathymetric estimate in our study region we utilised the new shipborne multibeam bathymetric data collected predominantly by a recent ITGC cruise, NBP19-02 (Fig. 1a) (Hogan et al., 2020). The topographic shift method was re-run with this multibeam data excluded from the constraining bathymetric dataset (Fig. 4a). The difference between the results with and without this test dataset (Figs. 4a and b) provides a snap-shot of the errors associated with our recovered bathymetry (Fig. 4c). In this region the mean error is -40 m, with a standard deviation of 100 m. We take this standard deviation to be representative of the expected error in our modelled bathymetry. This error is within the typical range for that quoted for gravity derived bathymetry, for example error estimates of ~60 m have been suggested in Greenland (An et al., 2019), while errors of up to ~160 m are suggested for the Larsen Ice Shelf where, unlike our study, no

account had been made for the underlying geology (Brisbourne et al., 2014;Cochran and Bell, 2012). The mean error we find indicates that the bathymetry constrained by the swath data is deeper than predicted by the gravity inversion alone, and hence that that there are geological features in this region distorting the recovered bathymetry. It is apparent that the largest errors are associated with higher frequency topography revealed by the new multibeam data. Such errors resulting from comparison of datasets with fundamentally different resolutions is to be expected, highlighting the need for multibeam bathymetry in regions where sub-kilometre-scale resolution of bathymetry is required. This is particularly relevant in areas where the seabed topography includes high amplitude variations at short wavelengths. In addition, this pattern of errors means that single seismic observations of cavity depth may not be ideal tie points for gravity inversions in rugged regions such as near Thwaites Glacier. A single such seismic measurement typically relies on a receiver array ~250 m long (Brisbourne et al., 2014) and hence could image a local high or low, biasing the wider gravity inversion.

### 4.2 Previous bathymetric estimates

Comparison between our topographic shift method and previous gravity inversions in this region show the broad sub-ice-shelf features are resolved by all methods, but differences in the detailed results are clear (SFig. 5). The OIB Level 3 data product (Tinto et al., 2011;Tinto and Bell, 2011) shows the largest discrepancies (SFigs. 5a,d and Figs. 3a,b), with our new inversion showing bathymetry 200 to 300 m shallower at the grounding line. This in part reflects the fact that the OIB bathymetric estimate was limited to using 2011 and older gravity data. In addition this bathymetric model relied on integrating the results of a series of 2D forward gravity models, incorporating observed bathymetry and radar-derived topography beneath the grounded ice. These gravity models did not factor in any regional trends in the gravity field, but rather corrected for a single DC shift at the outboard end of each profile, and modified the upper crustal density at the inland end of the profile to achieve a good fit. Un-modelled regional trends could, therefore, be a factor distorting the recovered bathymetry.

The Millan et al. (2017) inversion of bathymetry from gravity data shows the same general pattern of sub-ice-shelf bathymetry as our topographic shift method (SFig. 5b). However, differences are observed, most clearly beneath the inboard parts of the Dotson Ice Shelf (SFig. 3e). In addition significant undulation in the recovered bathymetry, not associated with any gravity signal are seen, for example from 100 to 120 km in Figure 3d. Such variability is indicative of artefacts due to the inversion approach. As our topographic shift method is different, and we incorporate additional new gravity, bathymetric and radar data, it is not immediately clear what the source of these discrepancies are. To independently asses the results of Millan et al., (2017) we compare their results with BedMachine Antarctica (Morlighem et al., 2020) (SFig. 4c) which used the same input data as Millan et al. (2017), and the gravity shift method previously applied to the Greenland margin (An et al., 2019). The key difference between the An et al., (2019) and Millan et al., (2017) methods is that the newer approach applies a variable rather than single DC shift to the gravity prior to inverting for the bathymetry. The residuals between the Millan et al., 2017 result and BedMachine Antarctica (the gravity shift method) (SFig. 4f) show a similar pattern to the residuals between the Millan et al., 2017 result and our topographic shift method (SFig. 4e). This indicates that the use of a single DC shift was a significant

issue in the older inversion (Millan et al., 2017) which may have led to an over-estimate of the depth of some near-shore features.

Comparing BedMachine Antarctica and our topographic shift results reveals that differences of over 250 m are still present (SFig. 5g). We suggest that these remaining differences reflect the additional multibeam bathymetric, ITGC radar and gravity data used in our topographic shift result. In addition the different bed topography onshore (OIB and ITGC line radar data here

vs. mass conservation in BedMachine Antarctica (Morlighem et al., 2020) and exclusion of sub-ice-shelf pinning points from our topographic shift result also likely contributed to the differences. For example topography with no associated gravity signal is seen in the BedMachine profile in Fig. 3d, indicating the method and tie points used introduced some artefacts. This highlights the need for caution when using -gravity-derived bathymetry and the value of high resolution gravity data with tight line spacing such as the integrated OIB/ITGC dataset, together with additional well-constrained and well distributed

observational tie points beneath the ice shelves, and around their margins.

## 4.3 Implications for the Amundsen Sea ice shelves

### 4.3.1 Pathways for water

The results of our new bathymetric estimates have significant implications for how we understand the pattern of cryospheric changes occurring in the Thwaites, Dotson and Crosson areas. Our primary observation confirms that the ice front in the centre

of Thwaites Glacier is directly and easily accessible to mCDW through a channel over 800 m deep beneath the Thwaites Eastern Ice Shelf and Thwaites Glacier Tongue (Millan et al., 2017;Tinto and Bell, 2011) (Fig. 2a). This trough is separated from an adjacent >1000 m deep trough by a ridge that is in places <150 m deep where the Eastern Ice Shelf and Thwaites Glacier Tongue were pinned. However, 700 to 800 m deep channels cut the ridge, linking the two troughs, and potentially facilitating lateral circulation beneath the ice shelves. Warm mCDW is dense, and could be filling the bathymetric depressions

and troughs on the continental shelf we observe, transporting heat from the global ocean to interact with ice shelves and contributing to ice sheet grounding line melting (Jenkins et al., 2010).

The Crosson Ice Shelf is underlain by bathymetry 300 to 500 m deep, shallower than the typical core of the mCDW (Assmann et al., 2013). A 700 – 1000 m deep channel is present flanking Bear Island (Figs. 2b and 3c), but its width of just 10 km suggests that the flux of mCDW may be less via this route. However, in some years the upper boundary of the mCDW can sit

around 400-600 m deep (Dutrieux et al., 2014;Jenkins et al., 2018), shallower than much of the bathymetry beneath the Crosson Ice Shelf, meaning mCDW could still access the inner Crosson cavity. The final ~30 km to the most recent grounding line of Smith Glacier are characterised by a cavity typically 100-200 m thick. As models indicate that reduced cavity thickness can supress strong oceanic circulation (Seroussi et al., 2017) this could limit the supply of mCDW water to the grounding line. The Dotson Ice Shelf is underlain by a broad cavity >800 m deep and is separated from the currently rapidly-changing grounding

line of the western branch of the Kohler Glacier by a sill 700-800 m deep (Fig. 3d). This sill may partially shield this grounding

line from oceanographically-driven change, as the bulk of the inflowing mCDW is mapped at a depth of ~800 m at the Dotson ice shelf margin (Miles et al., 2016).

### 4.3.2 Two ice shelf populations

A second key observation is that the ice shelves in areas which ungrounded since measurement of the 1993 grounding line are
all underlain by relatively thin cavities (Fig. 2b). Such thin cavity geometry in newly un-grounded regions is predicted by some fully coupled, ice-ocean numerical models of ice sheet retreat (Seroussi et al., 2017). These newly formed regions of floating ice (Fig. 1d) appear to be distinct from the wider, more established, ice shelf system which is underlain by both thick and thin cavities. This pattern is not simply a result of distance, and hence time since crossing the grounding line, as in places where the ice shelf has not advanced inland thick cavities are seen at the grounding line, for example west of Thwaites Glacier tongue,
and East of Pope Glacier (Fig. 2b). To consider the different ice shelf systems in more detail we plotted hydrostatic ice shelf draft against our recovered bathymetry (Fig. 5). This comparison utilised the 500 m resolution model of ice sheet draft derived from the REMA DEM (Section 2.3). Although the calculated ice shelf draft has higher resolution than the gravity derived bathymetry, the sampled bathymetry was interpolated smoothly between grid nodes providing a good estimate of how bathymetry changes at longer wavelengths across the region. Progressive down-sampling of the model of ice shelf draft did
not change the trend of the observed correlation, but reduced the number of points defining the trend (SFig. 6).

The older ice shelves, outboard from the 1993 grounding line, show limited correlation with the underlying bathymetry (Fig. 5a). This is expected given the shelves float passively over the underlying topography. Regionally the main control on the draft of these ice shelves is likely the depth of top of the mCDW, which drives enhanced basal melt. The fact that few of the older ice shelves have depths greater than 500 m is consistent with this hypothesis, as mCDW at depths of 400 to 800 m is observed
in oceanographic transects at the ice shelf edge (Miles et al., 2016;Dutrieux et al., 2014;Jenkins et al., 2018;Jacobs et al., 1996). The draft in newly established ice shelf areas shows an almost 1:1 relationship with the underlying bathymetry (Fig. 5b). The difference between the bed elevation and ice shelf draft suggests that these newly formed cavities are on average 112 m thick, with >95% being <~400 m thick. The rapid grounding line retreat which led to the formation of the post-1993 ice shelf sectors has been regarded as a harbinger of catastrophic collapse of the Amundsen Sea sector of the West Antarctic Ice Sheet through
geometric marine ice sheet instability, unconstrained by inland pinning points (Rignot et al., 2014). It has been suggested that basal melting driven by ingress of warmer mCDW could be a key factor facilitating this process (Milillo et al., 2019;Pritchard et al., 2012). Enhanced basal melting of up to 200 m per year has been calculated from satellite observations and OIB radar profiles over the new Thwaites Glacier ice shelves (Milillo et al., 2019), and rates of 50-70 m a$^{-1}$ have been observed close to the grounding line of Smith Glacier (Khazendar et al., 2016). However, our data indicate that the highest of these melt rates
must be restricted to the grounding line, as the newly formed cavity thickness typically does not exceed ~400 m, i.e. approximately two years of the most elevated melt rates, and 8 years at the lower end of the enhanced melt rates. We propose that the fast flowing ice is advected across the region of most enhanced melting, limiting subsequent thinning of the cavity.

This is in line with the suggestion of previous authors that where grounding line retreat is driven by melting, very high melt rates are likely focused at the grounding line (Lilien et al., 2019).

In the Smith Glacier region comparison of 2016 OIB radar with earlier radar data allows reconstruction of the spatial distribution of the most recent ice shelf thinning (Fig. 6). These direct observations confirm, as predicted from our cavity thickness estimate, that across much of the new ice shelf, thinning rates are relatively low, hence a relatively thin cavity can be maintained. However, they also reveal that the enhanced thinning rates of 50-70 m a$^{-1}$ beneath the inner shelf noted for the period 2002 to 2009 (Khazendar et al., 2016) have continued into the period 2009-2016. These high rates appear to be restricted

to the area where the base of the ice shelf is >1200 m deep. One possibility is that mCDW is penetrating to the grounding line and pooling at these depths. However, it is not clear to what extent this water would have been mixed and diluted during its passage through the <400 m thick sub-ice shelf cavity. Also, ice-shelf marginal weakening and consequent ice acceleration may have also contributed to the observed fast grounding-line retreat and thinning at the grounding line without the need for such extreme basal melting (Lilien et al., 2019).

The consistent presence of broad but vertically thin subglacial cavities appears to challenge a purely melt driven model of future ice sheet collapse, as access by warm water to the grounding line would be hampered by the thin cavity (Schoof, 2007). This physical limitation is supported by models for Pine Island Glacier margin, which indicated that cavities <200 m thick slowed the ingress of warm bottom water over topographic ridges (De Rydt et al., 2014). More complex fully coupled ocean-ice models also show the development of thin cavities and indicate that the associated weak circulation acts to slow grounding

line retreat relative to that predicted by an un-coupled model (Seroussi et al., 2017).

The contrast in cavity geometry and relationship to the underlying bathymetry of the pre and post 1993 ice shelf regions suggests that the recently un-grounded regions may not yet be in equilibrium with the wider glaciological and oceanic system. As such, they may play a significant, but as yet poorly understood role in controlling the future evolution of the ice sheet marginal system. The thin cavities in particular may act to slow future changes. Firstly, they place a fundamental limit onto

the amount of warm water which can flux beneath the glacier, and may also facilitate the tidally-driven turbulent flow mixing of water masses before they can reach the grounding line (Holland, 2008). In addition, the thin cavities which we observe are particularly sensitive to re-grounding on retrograde slopes, a negative feedback which would act to temporally re-stabilise a retreating ice sheet, a process which would be favoured by the observed rapid uplift due to glacial isostatic adjustment (Barletta et al., 2018). The process of grounding line re-advance appears to have occurred in the Western Kohler Glacier (Fig. 1a), where

the 2014 grounding line lies downstream of the 2012 grounding line (Rignot et al., 2014). Our observations of consistent thin cavities in newly un-grounded regions supports the results of coupled ocean-ice models confirming the necessity of such detailed modelling for predicting the evolution of the Thwaites Glacier system (Seroussi et al., 2017).

## 5. Conclusions

Airborne gravity provides a good first order estimate of sub-ice-shelf bathymetry. Despite the relatively high uncertainty (~100 m standard deviation) comparisons with different gravity inversion techniques, the location of ice shelf pinning points and new observational bathymetric data, indicate that the pattern of sub-ice-shelf bathymetry is well resolved.

Thwaites Glacier is connected to Pine Island Bay to its east by a major trough >800 m deep and 20 km wide. In contrast the grounding lines of the of Dotson and Crosson ice shelves are accessible through relatively narrow channels and thin sub shelf cavities.

In the Thwaites, Dotson and Crosson region, areas of ice shelf which developed before and after 1993 form distinct populations. The most recently un-grounded areas are underlain by thin cavities (average 112 m) where the ice shelf base closely tracks the underlying bed topography. We propose that these systems represent a transient phase of ice margin development, which may act to slow future changes, which is indicated but not fully captured in present models.

## Data availability

The new calculated bathymetry along with input topography and gravity grids is available from the UK Polar Data Centre (Jordan et al., 2020a). The associated ITGC line airborne gravity and magnetic data is available from the same source (Jordan et al., 2020b;Jordan et al., 2020c). Other data is from sources cited in the text.

## Author Contributions

All authors contributed to the discussion and production of the final manuscript.

Additional specific contributions include:

Tom Jordan – Gravity data processing, topographic shift bathymetric inversion and primary manuscript preparation.

David Porter – Airborne gravity and magnetic data collection.

Kirsty Tinto – Discussion of bathymetry inversion techniques.

Romain Millan – Discussion of gravity inversion and preparation of gravity shift bathymetric model.

Atsuhiro Muto - Discussion of gravity inversion and implications of ice sheet cavity findings.

Kelly Hogan – Preparation of multibeam swath bathymetry.

Rob Larter – Provision of ship-borne gravity and mutlibeam swath data.

Ali Graham – processing and preparation of mutlibeam swath bathymetry data on NBP19-02.

John Paden – led airborne radar system development, data processing, and ice bottom tracking.

**Competing Interests**

None

**Acknowledgements**

This work is a British Antarctic Survey (BAS) National Capability contribution to the International Thwaites Glacier Collaboration (ITGC), with additional support from the BAS Geology and Geophysics Team (TJ). Additional support for this
work is from Lamont-Doherty Earth Observatory NSF grant number NSF1842064 (DP, KT), and the THOR (KH, RL, AG, and Cruise NBP19 02), TARSAN (AM) and MELT (JP) projects, components of the International Thwaites Glacier Collaboration (ITGC). Support from National Science Foundation (NSF: Grant NSFPLR-NERC-1738942 for THOR, NSFPLR-NERC-1738992 for TARSAN, NSFPLR-NERC-1739003 for MELT) and Natural Environment Research Council (NERC: Grant NE/S006664/1 for THOR, NE/S006419/1 for TARSAN). This is paper number ITGC:009.

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

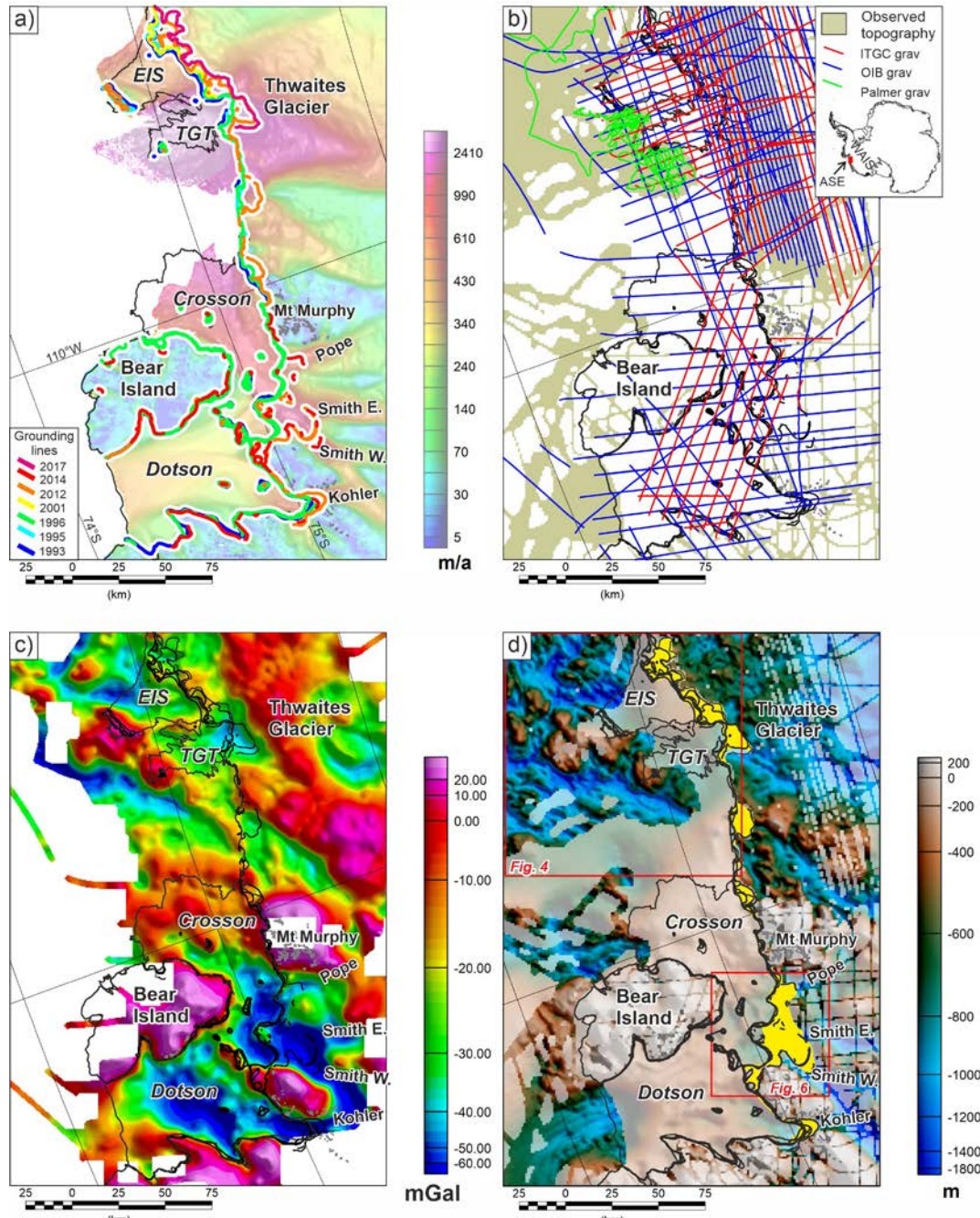

**Figure 1: Regional setting and input data. a) Ice velocity (Rignot et al., 2017) with InSAR grounding lines in colour (Rignot et al., 2014;Milillo et al., 2019). Grey lines rock exposures. Thwaites Glacier Tongue (TGT), Eastern ice Shelf (EIS). b) Line gravity data coverage, regions of known topography and inset showing Antarctic context. ASE is Amundsen Sea Embayment, WAIS West Antarctic Ice Sheet. c) Integrated free air gravity anomaly grid. d) Direct topographic observations (strong colours), onshore from Operation Ice Bridge (OIB) (Paden et al., 2010, updated 2018) and ITGC, and offshore from ship-borne swath coverage (Hogan et al. 2020). Pale colours show BEDMAP2 DEM (Fretwell et al., 2013). Yellow areas highlight post 1993 ice shelves. Red boxes locate figures 4 and 6.**


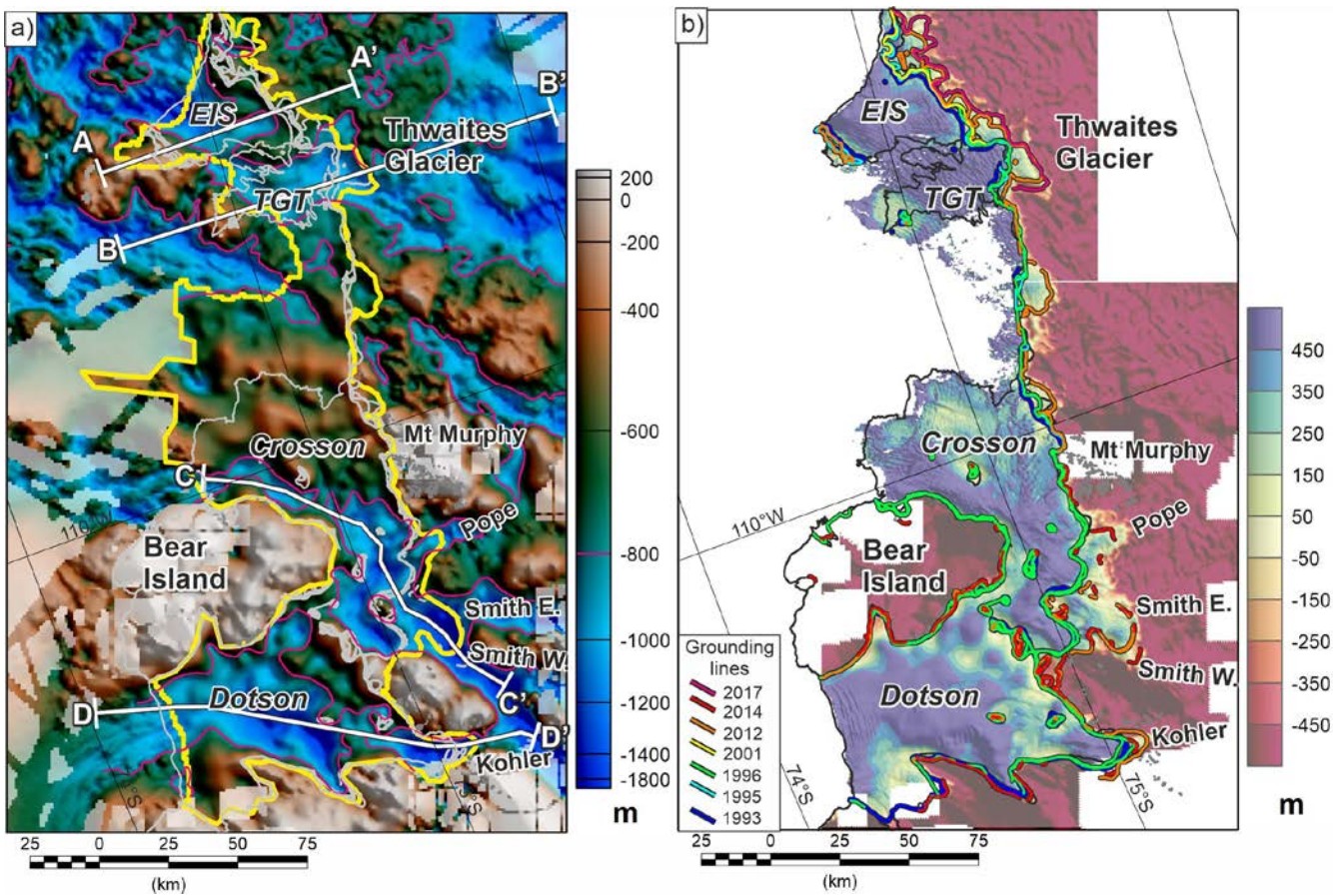


**Figure 2: New bathymetry and cavity maps. a) Final topography from terrain shift method. White lines A-D mark profiles in Fig. 3. Yellow outline encloses region constrained by gravity data. Pink line shows -800m depth contour. Light grey lines mark grounding lines and ice shelf edge. b) Sub-ice-shelf water column thickness based on the REMA DEM and an assumption of hydrostatic equilibrium. Regions where the ice sheet is predicted to be grounded show negative cavity thickness.**


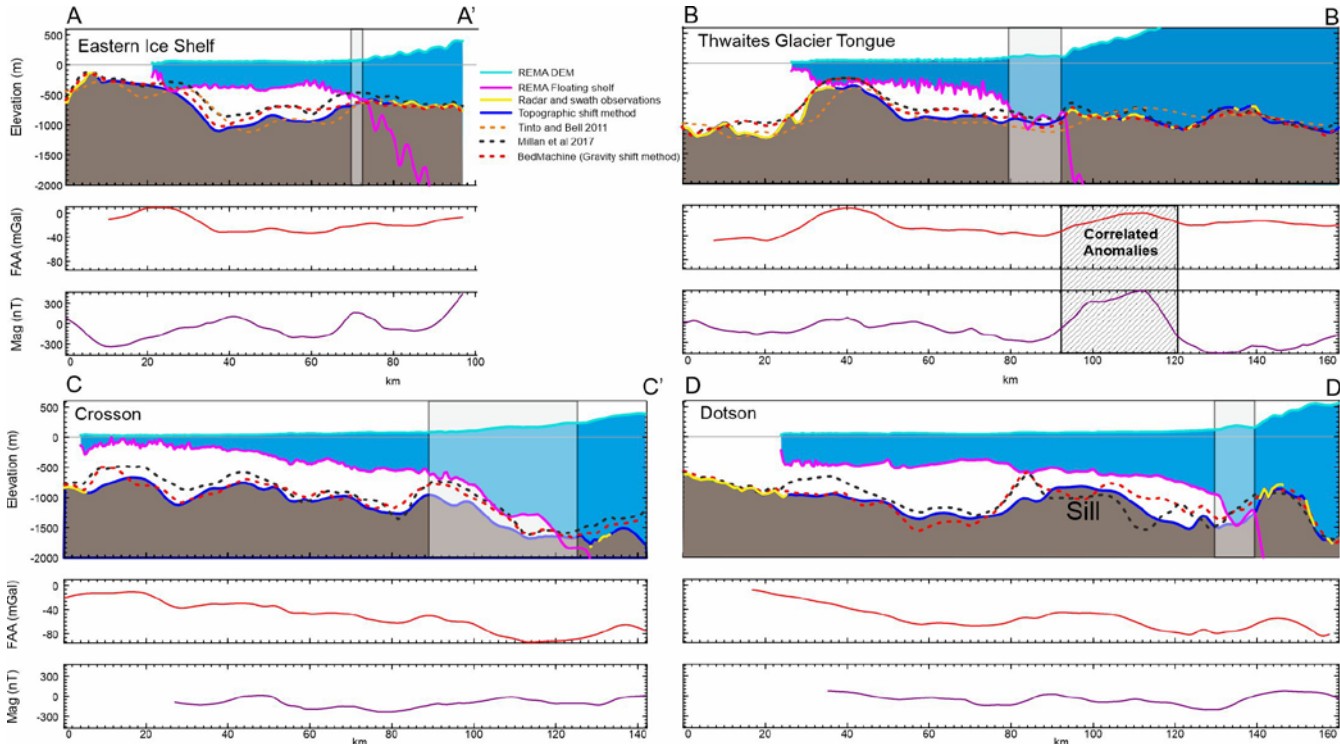

**Figure 3: Profiles across ice shelves. Upper panel shows ice surface from REMA DEM (Howat et al., 2019) and base of ice shelf calculated assuming hydrostatic equilibrium, together with gravity-derived bathymetric estimates. Second panel shows input free-air gravity anomaly. Third panel shows magnetic anomalies derived from ITGC survey data (Jordan et al., 2020b) and ADMAP2 (Golynsky et al., 2018). a) Thwaites Eastern Ice Shelf. b) Thwites Glacier Tongue. c) Crosson Ice Shelf. d) Dotson Ice Shelf. Note thin cavity in regions of ice sheet grounding line retreat since 1993 (grey boxes in upper panels). Also in (d) note >500 m bathymetric highs in BedMachine and Millan (2017) profiles not associated with a free air gravity anomaly, indicative of artefacts resulting from the inversion process.**


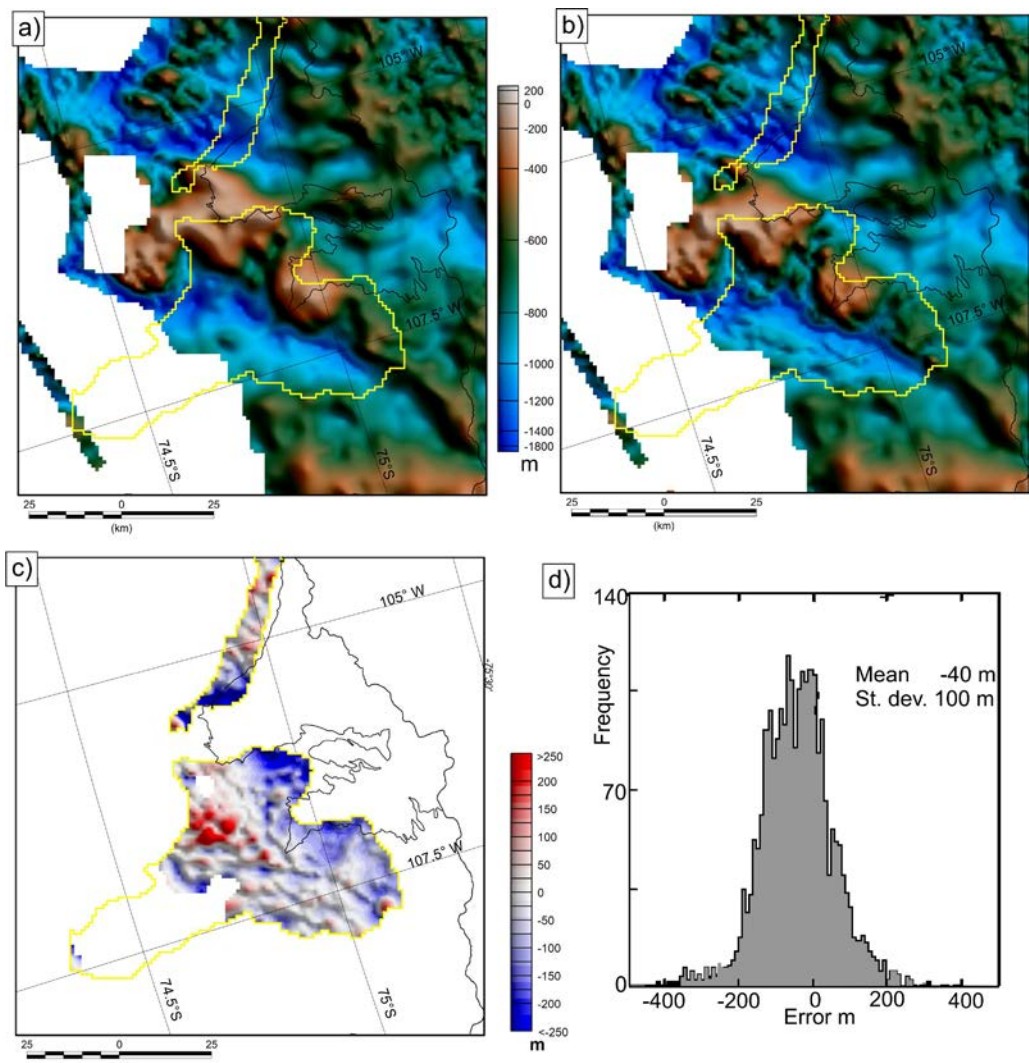


**Figure 4: Error estimates. a) Bathymetric estimate excluding bathymetric data from cruises NBP19-02 and JR294 (yellow outlines). b) Bathymetry including new multibeam data (as in Fig. 2a). c) Discrepancy in areas of additional data. d) Histogram of errors in areas of new multibeam constraint.**

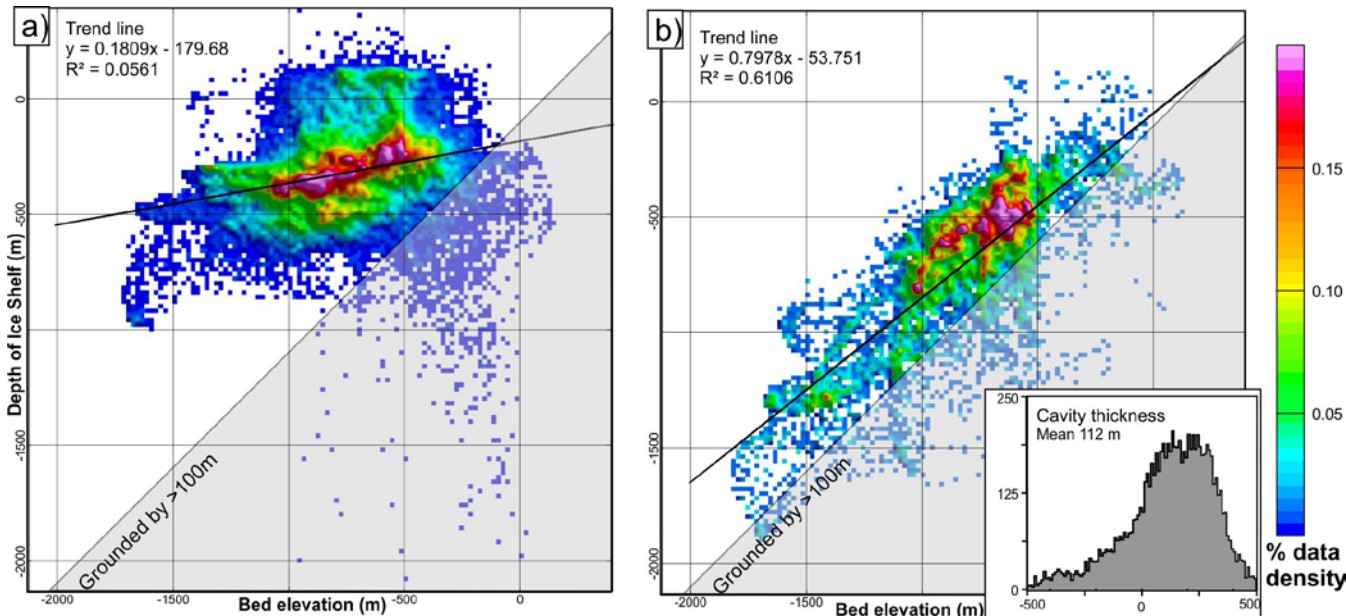

**Figure 5: Data density plot of hydrostatic ice shelf draft against sub-shelf bathymetry. Trend lines show best linear fit through data point cloud. a) Plot for ice shelves outboard of the 1993 grounding line. b) Plot for ice shelves formed by grounding line retreat since 1993, with inset showing histogram of cavity thickness beneath the areas of newly developed ice shelf. Note data where ice shelf depth is >0 m results from regions where the ice shelf surface elevation is less than the firn correction. Points which plot below the 1:1 line are theoretically grounded. However, errors in the gravity derived topography with a standard deviation of ~100 m are noted (Fig. 4d), hence some areas which appear shallowly grounded may in fact be floating. In addition uncertainties in grounding-line position and real pinning points within the areas designated as ice shelves contribute to the observed scatter of anomalous points.**

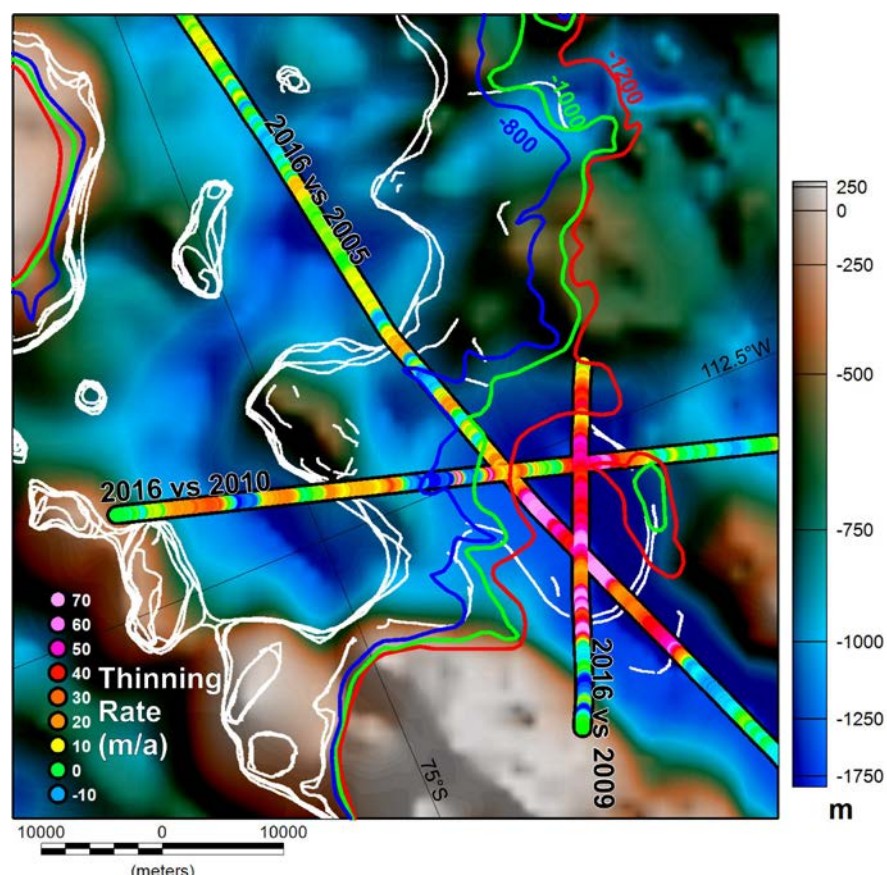


**Figure 6: Rate of Crosson Ice Shelf thinning determined by direct radio-echo sounding measurements from 2009, 2010, and 2016 OIB (Paden et al., 2010, updated 2018) and the 2005 AGASEA survey (Khazendar et al., 2016;Holt et al., 2006;Blankenship et al., 2012). Coloured contours show expected depth of base of floating ice shelf. White lines show INSAR derived grounding lines marking the front and back edges of the 'new' ice shelf (Rignot et al., 2014). For regional setting see Fig. 1d.**
