# Peer review of "New gravity-derived bathymetry for the Thwaites, Crosson and Dotson ice shelves revealing two ice shelf populations"

_The Cryosphere, 2019_

## Referee Comment (RC1) · Anonymous Referee #1 · 11 Feb 2020

Summary: The ice shelves play an important role in the stability of ice sheets through their buttressing effect. However, direct measurements could be difficult from different aspects, it is urgent to know what is happening underneath the ice shelves. So here, the authors present the improved topographic estimate underneath the Thwaites, Crosson and Dotson ice shelves (or the sub-ice-shelf cavity thickness) to help us how the warm ocean water access and interact with the glaciers' grounding lines. Overall, I have several questions about this manuscript. 1. According to Section2.2, the author mentioned they used a similar method in An et al., 2019, which refers here as the topographic shift method. Both of these techniques could take the variations in crustal thickness, sedimentary basins or intrusions into account, so is there a conclusion to

identify which method is better and why you choose the topographic shift method? 2. For Figure3, are the profiles across ice shelves, I am not sure this comparison makes sense. If I understand the material right, if the topographic shift method is constrained by Radar and swath observations, why the gravity shift method is not? In my opinion, both of these two methods should constrain by observations and inverted any other places where we don't have a direct measurement.

Minor: Text: Line237: typo Line250: format

---

## Referee Comment (RC2) · Anonymous Referee #2 · 29 Feb 2020

This paper presents new data and analysis and updates the sub-ice shelf bathymetry models of three major outlets of the West Antarctic Ice Sheet. As a technique paper this contribution is nice as it builds on previous work well and is convincing that the perhaps incremental improvements made here are worthwhile to do in bathymetry inversions. That being said, the discussion of error budget is lacking and while effort was made to compare the inversion to realistic observations to obtain a realistic error, this was only done in one relatively small area which I find inadequate for a general comment on uncertainty, especially considering the value is substantially lower than other similar work. The authors do a good job arguing that the basic bathymetry results should be an improvement over previous inversions of this area so it will be important for these

maps to be available for ongoing numerical ocean modeling work. However, I believe the release of the bathymetry to the modeling community is the main contribution of this paper; in its current form the scientific discussion reads rather speculatively with a somewhat awkward discussion of grounding line retreat that I find largely unnecessary.

Specific comments: Lines 54 to 56: The sentence referencing the Parker-Oldenburg method is misleading as it suggests that the problems discovered by the Cochran and Bell 2012 analysis that led to large disagreements between actual and inverted seafloor depth discussed in Brisbourne et al, 2014 were due to the algorithm used. The Parker-Oldenburg algorithm was never said to be the problem as there are many other more likely factors that may have contributed to the disagreement including platform speed, line-spacing/data coverage and resulting grid resolution, and, most importantly, the lack of explicit constraints on the geological forward model. To avoid misleading the readers, remove this discussion or replace with a full discussion of contributing factors, room permitting.

Line 92: The comparison in wavelengths between OIB and ITGC suggests an instrumentation difference; to clear this up please explain the improvement in resolution between the two campaigns –flight speed, elevation, instrumentation, etc.

Line 95: Please explain what you mean by ""will have little impact". If you mean that not upward continuing to a common elevation could introduce errors when you invert a gravity gridded field that assumes a common elevation then please state this is what you did. Although it seems right that +- 200m will have little impact, please add an estimate of the error introduced. Please also include an estimate of the error introduced for the 5% of the lines flown higher than 450 m and lower than 1500 m and refer to a map in the Supplement illustrating that those lines (or line segments) are not in areas where those introduced errors will impact your interpretations/results.

Line 100: What is the stated resolution and uncertainty of the GOCO3 gravity model? Please explain why the 2 mGal difference you observe more likely to be due to drift in

the marine system rather than a regional variation not captured in the GOCO3 model.

Line 181: Your error discussion currently highlights the 23 m contribution from crossover analysis and lack of geological knowledge. However, you have left out estimation of uncertainty due to platform speed, line spacing, and upward continuation. Either expand the discussion to including all sources of error in the budget or focus on the comparison with known bathymetry as you do later.

Line 200: Although I like your error estimation approach (comparing to known bathymetry), I don't think it is adequate to base your error for the whole survey region on only the multibeam area without at least showing that the errors are similar elsewhere; the multibeam area is less than 10% of your rather large survey area. This is additionally suspect as your 100 m error estimate is low compared to multiple other studies that quoted errors based on comparison with realistic bed data. This improvement in standard deviation is not expected considering that you are combining data from different platforms and instruments and your line spacing is coarse in many areas. Please present histograms for other areas to illustrate that both your mean and standard deviations are consistent where it matters –e.g. upstream of each grounding line. It may be helpful to compare your comparisons with known bathymetry to other studies that did something similar; the studies I'm aware of that also did this are: Brisbourne et al. 2014 (+-162 m), Greenbaum et al., 2015 (+-190 m), Hodgson et al. 2019 (+- 175 m).

Line 206: Typo: remove "there" after "where" Line 248: Please replace "typical shelf water" with something more descriptive.

Line 255: Please revise this sentence regarding MCDW supply. The supply of MCDW should be limited more by the depth of the shallowest bathymetry between source of the MCDW and the grounding line, not by the thickness of the water column near the grounding line. Profile C indicates a relatively shallow (500 m) sill which could reduce the supply of MCDW depending on the average thermocline depth which you refer to

as 400-600 m. Unless you meant something else by "limit the supply of mCDW". Later on line 319 you connect weak circulation with thin cavities, is that what you mean by limit the supply? If so, please connect this thought in both places.

Line 273-274: Your comment connecting the slight positive correlation to MCDW being forced onto shallow topography is very speculative and perhaps unnecessary; I recommend removing it otherwise please list other explanations for the correlation.

Line 320 to 330: It strikes me as an intuitive and even mundane result that more recently ungrounded ice shelf areas would have a tighter correlation with bathymetry than ice shelf areas that ungrounded previously. The discussion of this as it stands does not provide enough additional insight to convince me that the older shelf areas don't simply lose the correlation because they've just had more time to spread under their own weight and melt. It is also expected that recently ungrounded areas are the most likely to re-ground under a new flow regime or ocean conditions. I concede that I may have missed a subtle (or not so subtle) nuance, if so, please revise this discussion in a concise manner in your response otherwise I recommend shortening this section and moving the correlation plots to the supplement.

Figure 6 seems unnecessary when you can refer to the literature for this information. I recommend either moving it to the supplement or at least stacking them vertically and placing them next to the thinning map in Figure 7 to save space.

References: Please add standard indentation to improve readability for the next revision.

---

## Author Comment (AC1) · 15 May 2020

**Response to reviewers and editorial comments on "New-gravity derived bathymetry for the Thwaites, Crosson and Dotson ice shelves reveals two ice shelf populations".**

Reviewers comments in black, author responses in blue.

**Reviewer 1:**

Summary: The ice shelves play an important role in the stability of ice sheets through their buttressing effect. However, direct measurements could be difficult from different aspects, it is urgent to know what is happening underneath the ice shelves. So here, the authors present the improved topographic estimate underneath the Thwaites, Crosson and Dotson ice shelves (or the sub-ice-shelf cavity thickness) to help us how the warm ocean water access and interact with the glaciers' grounding lines.

Overall, I have several questions about this manuscript:

1. According to Section2.2, the author mentioned they used a similar method in An et al., 2019, which refers here as the topographic shift method. Both of these techniques could take the variations in crustal thickness, sedimentary basins or intrusions into account, so is there a conclusion to identify which method is better and why you choose the topographic shift method?

2. For Figure3, are the profiles across ice shelves, I am not sure this comparison makes sense. If I understand the material right, if the topographic shift method is constrained by Radar and swath observations, why the gravity shift method is not? In my opinion, both of these two methods should constrain by observations and inverted any other places where we don't have a direct measurement.

Minor: Text: Line237: typo Line250: format

We thank reviewer 1 for their comments on our manuscript. Our response to the key points raised is below:

- 1. We do not think that either the topographic shift or gravity shift method of An et al is intrinsically better. Both use topographic observations to constrain the final gravity derived topography, and both use interpolation of the error field between the known topographic points to generate an optimum correction. The use of constraining data to remove long wavelength trends induced by regional geology is the key point about both these methods, and where they are an improvement on the earlier method in this area which used a single DC shift. Our choice of the topographic shift method was simply driven by easy access to software.
- 2. Both the gravity shift method implemented in BedMachine (following the method of An et al) and the topography shift method we apply are constrained by observational data. However, the topographic shift approach is constrained by additional new swath and radar data collected for ITGC. The gravity shift method as implemented in BedMachine used the same, but more limited, input data as the older Millan 2017 method. This allowed us to show the key improvement from having a regionally varying correction, rather than the simple DC shift method. The topographic shift method we apply also excludes sub-ice shelf pinning points, which in some places appear to give rise to artefacts evident as topography where there is no gravity signal (Fig. 3d).

**Referee #2**

This paper presents new data and analysis and updates the sub-ice shelf bathymetry models of three major outlets of the West Antarctic Ice Sheet. As a technique paper this contribution is nice as it builds on previous

work well and is convincing that the perhaps incremental improvements made here are worthwhile to do in bathymetry inversions.

That being said, the discussion of error budget is lacking and while effort was made to compare the inversion to realistic observations to obtain a realistic error, this was only done in one relatively small area which I find inadequate for a general comment on uncertainty, especially considering the value is substantially lower than other similar work. The authors do a good job arguing that the basic bathymetry results should be an improvement over previous inversions of this area so it will be important for these maps to be available for ongoing numerical ocean modeling work. However, I believe the release of the bathymetry to the modeling community is the main contribution of this paper; in its current form the scientific discussion reads rather speculatively with a somewhat awkward discussion of grounding line retreat that I find largely unnecessary.

We thank reviewer 2 for their comments and provide a detailed response on a point by point basis below.

**Specific comments:**

Lines 54 to 56: The sentence referencing the Parker-Oldenburg method is misleading as it suggests that the problems discovered by the Cochran and Bell 2012 analysis that led to large disagreements between actual and inverted seafloor depth discussed in Brisbourne et al, 2014 were due to the algorithm used. The Parker-Oldenburg algorithm was never said to be the problem as there are many other more likely factors that may have contributed to the disagreement including platform speed, line-spacing/data coverage and resulting grid resolution, and, most importantly, the lack of explicit constraints on the geological forward model. To avoid misleading the readers, remove this discussion or replace with a full discussion of contributing factors, room permitting.

We agree that the Parker-Oldenburg algorithm was never considered the problem. We will therefore re-worded this section to clarify the fact that the problem lies in transformation of gravity signals directly into equivalent topography, and that such errors in the Larsen case-study were most likely to be due to the lack of explicit constraints on the underlying geology.

An overview of previous methods is an important introduction to our paper. We have therefore feel this section should be retained.

The additional point on the impact of data resolution, a function of platform speed, line spacing, data coverage and altitude is correct. We will add the following text later in the manuscript: "In addition to quantifying the errors it is important to note that the resolution of the bathymetry recovered from gravity data is limited by the wavelengths resolved by the gravity systems and the survey line spacing. For this

study the gravity systems resolved minimum wavelengths of 5 to 10 km and a minimum line spacing of ~5 km is achieved outboard of Thwaites Glacier, while a minimum line spacing of ~7.5 km was achieved over the Dotson and Crosson Ice Shelves. This study therefore only recovered bathymetric features with a wavelength of ~5 km and upwards".

Line 92: The comparison in wavelengths between OIB and ITGC suggests an instrumentation difference; to clear this up please explain the improvement in resolution between the two campaigns –flight speed, elevation, instrumentation, etc.

The reviewer is correct, the OIB and ITGC use different platforms and gravity instruments and this will be clarified in an updated version of the text. We note that the ITGC data was collected on a twin otter platform which flies approximately half the speed of the OIB aircraft, so resolution of shorter wavelengths may be expected.

Line 95: Please explain what you mean by ""will have little impact". If you mean that not upward continuing to a common elevation could introduce errors when you invert a gravity gridded field that assumes a common

elevation then please state this is what you did. Although it seems right that +-200m will have little impact, please add an estimate of the error introduced. Please also include an estimate of the error introduced for the 5% of the lines flown higher than 450 m and lower than 1500 m and refer to a map in the Supplement illustrating that those lines (or line segments) are not in areas where those introduced errors will impact your interpretations/results.

By little impact we mean that continuation by +/- 200 m will change the peak amplitude of the observed gravity anomalies by up to  $\sim$ +/-1 mGal. This would equate to  $\sim$ +/- 14 m. variation in bathymetry which is well below any reasonable error for the recovered bathymetry. This point will be amended in any revised text.

A supplementary figure showing the range to ground on all gravity flight lines will be added to the text (and at the end of this document). It is apparent that one flight outboard from Thwaites glacier is flown ~200 m lower than the others. This may add a little extra resolution to the recovered bathymetry, but as noted above, not much. Over the Dotson/Crosson system there is one section of an OIB flight which is notably higher. This may bias the recovered bathymetry to be smoother than it really is, however, as noted in the text, downward continuation of this line would potentially introduce errors resulting from continuation procedure. We also note that if these line sections were contributing appreciable anomalous high or low frequency signals they would be apparent in the hill-shaded free air gravity anomaly field (Fig. 1c), which they are not.

Line 100: What is the stated resolution and uncertainty of the GOCO3 gravity model? Please explain why the 2 mGal difference you observe more likely to be due to drift in the marine system rather than a regional variation not captured in the GOCO3 model.

The GOCO3 field is accurate to degree and order 250 (~160 km). We can-not rule out shorter wavelength geoid variability, or alternatively, that ice mass loss, coupled with GIA has locally altered the Geoid-Ellipsoid separation accounting for the residual 2 mGal error. The text will be updated on this point. However, we note that none of these issues will impact on recovery of the local bathymetry.

Line 181: Your error discussion currently highlights the 23 m contribution from crossover analysis and lack of geological knowledge. However, you have left out estimation of uncertainty due to platform speed, line spacing, and upward continuation. Either expand the discussion to including all sources of error in the budget or focus on the comparison with known bathymetry as you do later.

We believe the point the reviewer is making here relates to the intrinsic issue of the limited resolution of gravity derived bathymetry, which are a function of platform performance, speed, line spacing and flight elevation. This point on resolution is addressed above.

Line 200: Although I like your error estimation approach (comparing to known bathymetry), I don't think it is adequate to base your error for the whole survey region on only the multibeam area without at least showing that the errors are similar elsewhere; the multibeam area is less than 10% of your rather large survey area. This is additionally suspect as your 100 m error estimate is low compared to multiple other studies that quoted errors based on comparison with realistic bed data. This improvement in standard deviation is not expected considering that you are combining data from different platforms and instruments and your line spacing is coarse in many areas. Please present histograms for other areas to illustrate that both your mean and standard deviations are comparisons with known bathymetry to other studies that did something similar; the studies I'm aware of that also did this are: Brisbourne et al. 2014 (+-162 m), Greenbaum et al., 2015 (+-190 m), Hodgson et al. 2019 (+-175 m).

As noted in the text deriving error estimates for gravity derived bathymetry is challenging. If we had observations to do the error analysis we wouldn't need to do the inversion.

On the specific suggestion to perform similar analysis (exclude observational data – re-compute – and generate new error histograms) upstream of the Thwaites grounding line. This region unfortunately contains a significant geological structure which biases the gravity field. Any error estimate using the above approach would therefore suggest an error of several hundred meters, which the swath comparison suggests is not representative of the wider survey. We can-not categorically rule out the presence of similar geological bodies elsewhere in the survey area, however, the onshore structure has a distinct correlated high amplitude magnetic signature (Fig. 3b), which we do not see elsewhere beneath the ice shelves. The submarine environment that was used for the uncertainty comparison is also likely more representative of the submarine environment of the rest of the survey region than comparison with subglacial or subaerial exposed topography.

We note that an error of +- 100 m is not unreasonably low. For example An et al., 2019 estimated the error of the analogous gravity shift method to be ~60 m along the Greenland margin, while Tinto et al reported an estimated error of ~68 m for the Ross Ice Shelf area, excluding sensitivity to unconstrained geological variations. It is unsurprising that the Larsen Ice shelf gave higher errors, as no robust account was made for the geology of the region. In the case of the Hodgson et al. 2019 paper the 175 m error was before adjustment to the controlling seismic, swath and radar data, and should be considered a worst-case error for an unconstrained inversion of gravity data. The true error in that study will by definition be less than the suggested 175 m. Hodgson et al. 2019 suggest an error estimate of ~100 m is appropriate, based on how accurately the gravity derived bathymetry predicts ice shelf grounding depths.

Line 206: Typo: remove "there" after "where" Amended Line 248: Please replace "typical shelf water" with something more descriptive. Re-written.

Line 255: Please revise this sentence regarding MCDW supply. The supply of MCDW should be limited more by the depth of the shallowest bathymetry between source of the MCDW and the grounding line, not by the thickness of the water column near the grounding line. Profile C indicates a relatively shallow (500 m) sill which could reduce the supply of MCDW depending on the average thermocline depth which you refer to as 400-600 m. Unless you meant something else by "limit the supply of mCDW". Later on line 319 you connect weak circulation with thin cavities, is that what you mean by TCD limit the supply? If so, please connect this thought in both places.

We acknowledge that shallow bathymetry is the most obvious limit mCDW flow, however, in this case it is the cavity thickness that we suggest could limit flow. We will make this point more explicitly in the revised text and, as suggested, will better link this sentence to the part later in the paper.

Line 273-274: Your comment connecting the slight positive correlation to MCDW being forced onto shallow topography is very speculative and perhaps unnecessary; I recommend removing it otherwise please list other explanations for the correlation.

**OK we have removed this speculation.**

Line 320 to 330: It strikes me as an intuitive and even mundane result that more recently ungrounded ice shelf areas would have a tighter correlation with bathymetry than ice shelf areas that ungrounded previously. The discussion of this as it stands does not provide enough additional insight to convince me that the older shelf areas don't simply lose the correlation because they've just had more time to spread under their own weight and

melt. It is also expected that recently ungrounded areas are the most likely to re-ground under a new flow regime or ocean conditions. I concede that I may have missed a subtle (or not so subtle) nuance, if so, please revise this discussion in a concise manner in your response otherwise I recommend shortening this section and moving the correlation plots to the supplement.

We acknowledge that this section was overly complicated and will shorten it to make it more concise (Including removing Figure 6). However, while we agree that the observation of thin new sub-shelf cavities may be intuitive, we do not believe it is mundane for two reasons.

1: The very large areas (>30 km wide) covered by thin cavities indicate that extremely high melt rates required to drive retreat are focused at the grounding line, and not a widely dispersed phenomenon. This is a result which must be predicted by any future model of ice shelf evolution.

2: The basal profile of an ice shelf in dynamic equilibrium with the ocean in this region would rise steeply seaward of the grounding line and continue rising at least until it reaches the mCDW thermocline depth. We show that this is not the case for the recently evolved ice shelves, hence using a simple equilibrium model would miss important details of the system evolution.

Figure 6 seems unnecessary when you can refer to the literature for this information. I recommend either moving it to the supplement or at least stacking them vertically and placing them next to the thinning map in Figure 7 to save space.

**Figure will be removed**

References: Please add standard indentation to improve readability for the next revision.

**Format will be updated.**

**Editor's note:**

Dear Dr. Jordan et al.,

I've read through your initial submission and consider it suitable to send out to review. However, if the referees don't bring it up, I will strongly recommend the following:

- While published only recently, the authors are certainly aware of BedMachine Antarctica (Morlighem et al, 2019, Nature Geoscience), which is a significant advance over Bedmap2. The manuscript as it stands is immediately dated by a comparison with Bedmap2 in Figure 1, and some of the discussion of the differences with Milan et al. (2017) partly stems from their use of MC onshore, which I assume was also incorporated in Morlighem et al. (2019). I encourage the authors to switch over to using BedMachine Antarctica absent a contraindicating argument, which will extend the shelf life of this manuscript.

We recognise that BedMachine Antarctica was a significant advance on Bedmap2 in many respects. BedMachine Antarctica included bathymetry derived using the gravity shift method. We will therefore replaced our comparisons between the gravity shift results with comparisons to BedMachine Antarctica (see supplementary figure 2 below).

However, we note that the mass conservation approach does make assumptions and can change 'real' topography observed by radar. By constraining our recovered bathymetry only with observed radar derived topography, or swath bathymetry, we ensure our inversion is independent of as many additional assumptions as possible. Given the very good radar coverage in the Thwaites glacier region the use of mass conservation rather

than the 'real' data should have relatively little impact. It is interesting to note that BedMachine shows a consistent offset of ~34 m for the onshore part of the Thwaites catchment compared to our result which is derived simply from the radar bed pick and geoid correction. The origin of this offset is not clear to the authors, but further investigation of this on-shore observation is beyond the scope of this paper.

In addition our constraining compilation includes new radar and multibeam bathymetry data which were collected and processed after BedMachine Antarctica was produced. Although the new radar data is a relatively minor component, in regions such as Bear Island which were sparsely covered, they provide important constraints. By including these new data we ensure the best possible constraint on our predicted sub-ice shelf bathymetry.

- More exposition and emphasis on the two-population concept that you introduce in the title. Figure 5 is quite interesting and compelling support of the concept, but I found the two panels in Figure 6 difficult to distinguish from each other.

In line with your suggestion and that of reviewer 2 we have made the section on the two ice shelf populations more concise and removed Figure 6, as we did not feel it added enough to the discussion.

- In several figures, reconsider the green-to-red grounding-line history overlain on green-to-red topography.

We have amended the colour-scales of the underlying datasets on these figures so that the grounding-lines stand out more clearly.

Thanks for submitting your work to The Cryosphere,

Joe MacGregor NASA/GSFC

**Supplementary figures**

---

## Author Response (AR2)

To the Editor

We thank the editor for their final comments. They were useful, and included a point on resolution we had not fully considered. Although this point is important, and is now discussed in the text, we note that it does not change our results, or impact on our discussion. Our detailed responses to all the points raised are laid out below.

Regards.

Tom Jordan

Thanks again for your submission to The Cryosphere. I've reviewed your revised MS and consider it responsive to the reviewer's comments. The MS is substantially improved and the potential significance of the two ice-shelf populations is now much clearer. Below I only have a few minor technical corrections for you to consider.

17: Glacier OK amended.

23: For the abstract, be more specific than "new", because what is meant could easily be unclear to the unfamiliar reader. Perhaps something like "recently formed" or "recently floated". New is too vague and could imply that a new ice shelf formed from ice flux>(melting+calving) at a grounding line, rather than expanded inland. OK went with "recently floated".

19: tens of kilometers OK amended.

94-96: Clarify platform used and speed, in a similar to OIB above. Amended as suggested.

99: Good spot for a paragraph break? Paragraph breaks now at L97 & L104 to improve flow.

288: The discussion surrounding the two ice-shelf populations is much improved, but both earlier in the MS and here it's not so clear what the spatial resolution of this intercomparison is. I'm assuming it's the 1 km grid size, but is that appropriate for this intercomparison given the resolving power of airborne gravity? I kept expecting this issue to be discussed, but never saw it. The improved Figure 5 gives confidence in the significance of the difference, but I suspect it's less data-rich than implied here.

This is a very good point, which we had not fully considered. We utilised the 500 m ice shelf keel depth raster derived from the REMA DEM as the starting point for this inter-comparison. This is now also noted in the methods section as it impacts on the calculated cavity thickness map as well. The REMA derived DEM obviously has higher spatial resolution than the gravity derived bathymetry. However, as the depth of the bathymetry is interpolated smoothly between grid nodes for the comparison it will give a good estimate of how bathymetry changes across the region. Sampling this regional trend multiple times should allow long wavelength co-variance between the two signals to be better defined, although high frequency features from the ice shelf keel depth will add apparent noise.

To test the impact of the over sampling due to using a higher resolution ice shelf DEM we now additionally compare the results for ice shelf keel-depth rasters re-gridded at 1 km and 5 km cell size (New supplementary figure 6). This analysis shows that although the number of comparisons decreases by two orders of magnitude between the end member cases, the distinct approximately 1:1 trend seen for the new ice shelves remains clear.

Figure 5: Include number of values (n), even better (given comment above), number of independent samples if possible? N values have been added to the new supplementary figure.

[revised manuscript text omitted]